# A System Dynamics and Participatory Action Research Approach to Promote Healthy Living and a Healthy Weight among 10–14-Year-Old Adolescents in Amsterdam: The LIKE Programme

**DOI:** 10.3390/ijerph17144928

**Published:** 2020-07-08

**Authors:** Wilma E. Waterlander, Angie Luna Pinzon, Arnoud Verhoeff, Karen den Hertog, Teatske Altenburg, Coosje Dijkstra, Jutka Halberstadt, Roel Hermans, Carry Renders, Jacob Seidell, Amika Singh, Manou Anselma, Vincent Busch, Helga Emke, Emma van den Eynde, Lieke van Houtum, Wilma J. Nusselder, Meredith Overman, Susan van de Vlasakker, Tanja Vrijkotte, Erica van den Akker, Stef P. J. Kremers, Mai Chinapaw, Karien Stronks

**Affiliations:** 1Department of Public and Occupational Health, Amsterdam Public Health Research Institute, Amsterdam UMC, University of Amsterdam, Meibergdreef 9, 1105 AZ Amsterdam, The Netherlands; a.c.LunaPinzon@amsterdamumc.nl (A.L.P.); t.vrijkotte@amsterdamumc.nl (T.V.); k.stronks@amsterdamumc.nl (K.S.); 2Public Health Service Amsterdam, Sarphati Amsterdam, 1018 WT Amsterdam, The Netherlands; averhoeff@ggd.amsterdam.nl (A.V.); vbusch@ggd.amsterdam.nl (V.B.); lvhoutum@ggd.amsterdam.nl (L.v.H.); svdvlasakker@ggd.amsterdam.nl (S.v.d.V.); 3Department of Sociology, University of Amsterdam, 1018 WV Amsterdam, The Netherlands; 4Amsterdam Healthy Weight Programme, Public Health Service Amsterdam, 1018 WT Amsterdam, The Netherlands; Karen.den.Hertog@amsterdam.nl; 5Department of Public and Occupational Health, Amsterdam Public Health Research Institute, Amsterdam UMC, Vrije Universiteit Amsterdam, Van der Boechorststraat 7, NL-1081 BT Amsterdam, The Netherlands; t.altenburg@amsterdamumc.nl (T.A.); a.singh@amsterdamumc.nl (A.S.); m.anselma@amsterdamumc.nl (M.A.); h.emke@amsterdamumc.nl (H.E.); m.chinapaw@amsterdamumc.nl (M.C.); 6Department of Health Sciences, Amsterdam Public Health Research Institute, Vrije Universiteit Amsterdam, 1081 HV Amsterdam, The Netherlands; coosje.dijkstra@vu.nl (C.D.); j.halberstadt@vu.nl (J.H.); carry.renders@vu.nl (C.R.); j.c.seidell@vu.nl (J.S.); 7Department of Health Promotion, NUTRIM School of Nutrition and Translational Research in Metabolism, Maastricht University, 6229 ER Maastricht, The Netherlands; r.hermans@maastrichtuniversity.nl (R.H.); m.overman@maastrichtuniversity.nl (M.O.); s.kremers@maastrichtuniversity.nl (S.P.J.K.); 8Department of Pediatric Endocrinology, Erasmus MC, University Medical Center, 3025 GD Rotterdam, The Netherlands; E.vandeneynde@erasmusmc.nl (E.v.d.E.); e.l.t.vandenakker@erasmusmc.nl (E.v.d.A.); 9Department of Public Health, Erasmus MC, University Medical Center Rotterdam, 3015 CN Rotterdam, The Netherlands; w.nusselder@erasmusmc.nl

**Keywords:** systems dynamics, participatory action research, Amsterdam Healthy Weight Programme, co-design, overweight and obesity, adolescents, complex systems

## Abstract

This paper describes the design of the LIKE programme, which aims to tackle the complex problem of childhood overweight and obesity in 10–14-year-old adolescents using a systems dynamics and participatory approach. The LIKE programme focuses on the transition period from 10-years-old to teenager and was implemented in collaboration with the Amsterdam Healthy Weight Programme (AHWP) in Amsterdam-East, the Netherlands. The aim is to develop, implement and evaluate an integrated action programme at the levels of family, school, neighbourhood, health care and city. Following the principles of Participatory Action Research (PAR), we worked with our population and societal stakeholders as co-creators. Applying a system lens, we first obtained a dynamic picture of the pre-existing systems that shape adolescents’ behaviour relating to diet, physical activity, sleep and screen use. The subsequent action programme development was dynamic and adaptive, including quick actions focusing on system elements (quick evaluating, adapting and possibly catalysing further action) and more long-term actions focusing on system goals and/or paradigm change. The programme is supported by a developmental systems evaluation and the Intervention Level Framework, supplemented with routinely collected data on weight status and health behaviour change over a period of five years. In the coming years, we will report how this approach has worked to provide a robust understanding of the programme’s effectiveness within a complex dynamic system. In the meantime, we hope our study design serves as a source of inspiration for other public health intervention studies in complex systems.

## 1. Background

The prevention of childhood overweight and obesity and related determinants, including poor diet, poor sleep, excessive screen use and insufficient physical activity, are considered complex health problems. The drivers are multiple, diverse, and dynamic, ranging from biological factors and personal behaviours to aspects of the physical, economic, sociocultural and political environments [1,2]. This particularly applies to adolescents as they go through major physical, emotional and social changes. Addressing these complex problems transcends organisational and sectoral levels, where the various actors involved have different perceptions about the nature of the problem and the desirability and directions of the solutions [3,4].

Given the complexity of causation, it is now widely accepted in public health that solutions for overweight and obesity cannot solely be found in individual-level actions (e.g., health education), and should additionally include environmental determinants such as the accessibility of healthy food [5,6]. However, at the environmental level, we also see that interventions do not always achieve the anticipated effects [7]. An important reason for this lack of effect is that interventions generally have a narrow public health focus and target a limited number of determinants at a single environmental level, such as a school, instead of more systemic actions including multiple levels, which are required to achieve a meaningful effect on a complex problem such as overweight [7]. For example, the effectiveness of preventive school-based programs aimed at improving the food and beverage availability in canteens and vending machines *in* schools may be impacted by the presence of multiple fast food outlets *around* these schools [8]. Moreover, sustainable implementation is often lacking because of difficulties associated with ownership among stakeholders and competing policy interests (e.g., economic versus public health which we see around implementation of a soft drink tax), as well as with the need for targeted intervention strategies for diverse populations and contexts in real-life settings [9,10].

In view of the increasing complexity of interventions, as well as the context in which they are implemented, there is growing consensus that shifts in the public health research paradigm are needed. A first shift is *systems thinking*. Systems thinking refers to the notion that we should understand the broader system if we want to successfully address complex problems such as childhood overweight and obesity. Systems thinking goes beyond an ecological perspective that aims to direct an intervention at multiple levels (such as child, family, school, community and health care), and instead recognises each level as a system that follows the principles of system dynamics including linkages, relationships, feedback loops and interactions among the system parts, as well as the dynamics between levels [7,11]. For example, healthy schools might set a precedent for levels outside the school, and therefore change social norms in families and communities, or might yield the opposite effect (by conducting extra unhealthy behaviour outside school). The outcomes that result from these interactions, such as increasing rates of overweight and obesity, are considered ‘emergent properties’ of the system, which, by definition, cannot be fully understood by examining the individual system parts [11].

Second, there is a growing recognition for the need to include target populations in the development and implementation of interventions. Interventions that have been co-created with relevant stakeholders are more likely to meet their local theories of aetiology and change as well as local practices, and are therefore more likely to achieve effects [12]. Participatory Action Research (PAR) allows accommodating different perspectives on the aetiology of and solutions for public health problems, which is considered an important aspect of complex problems [13].

In parallel with the acknowledgement of the need for systems thinking and PAR when developing public health interventions, the methodology of evaluation also increasingly accounts for these paradigm shifts. Whereas the randomized controlled trial (RCT) is traditionally seen as the ‘gold standard’ in evaluation research, it is merely suited to evaluate single and multicomponent interventions in a controlled setting. Process evaluations are a good complementary method to RCTs to provide further insight into the working mechanisms of interventions, the intervention context, and the implementation process [14]. RCT’s are, however, not well suited for studying interventions in complex systems, particularly because there is no clear distinction of intervention effect on the one hand, and context and implementation at the other. Instead, there are multiple, interacting factors that are adaptive and dynamic over time and solutions are about changing the system dynamics in favourable ways [15], rather than implementing a demarcated intervention. The combination with participatory action research poses further challenges for the methodology of evaluation research, e.g., because it makes the relationship between researchers and the target population part of the context in which the intervention is being developed and evaluated. Also, the reflexive learning and informed iterative adaptation of interventions which characterise participatory action research implies that the intervention evolves over time, meaning that single intervention components cannot be clearly defined, let alone be tested in a controlled pre-post design.

Despite growing popularity, there still is relatively little guidance on how to best design and evaluate co-designed public health interventions in complex systems [16,17]. A systematic review on systems thinking in public health showed that the vast majority of publications in this area were opinion pieces/commentaries [18], and a second review found that most studies took a multicomponent approach rather than complex systems [19]. There are some relevant examples of original studies in the recent literature [20], including The Whole of Systems Trial of Prevention Strategies for Childhood Obesity (WHOSTOPS), which uses community-based system dynamics to mobilize community action for childhood obesity prevention in Australia [17,21]. LIKE builds on the methodology described in that study but does not specifically aim to mobilize community action. Instead, LIKE aims to create and measure changes in various system levels (which could or could not include community action). Here, we also aim to enhance the trial evaluation design still used in most complex childhood obesity intervention studies [22] by adopting a systems dynamics evaluation approach.

### Aims & Objectives

The purpose of this paper was to describe the design of the LIKE (Lifestyle Innovations based on youths’ Knowledge and Experience) programme, which aims to tackle the complex problem of childhood overweight and obesity in the Netherlands in 10–14-year-old adolescents in a lower socioeconomic, ethnically diverse group, using a systems dynamics approach and participatory action research. LIKE is a five-year programme (mid 2018–mid 2022), implemented in three districts in Amsterdam East with around 30,000 inhabitants. The LIKE programme is being developed and implemented within the Amsterdam Healthy Weight Programme (*AHWP*). This municipality-led programme aims to reduce childhood overweight and obesity in Amsterdam in a long-term timeframe (2013–2033) [23]. The AHWP recognizes overweight and obesity as a complex problem and focuses on transforming the local environment (political, physical, social, health education and care) to provide optimal conditions for healthy patterns of physical activity, sedentary behaviour, sleep and diet. The partnership with AHWP offers a solid foundation for the development and implementation, as well as the evaluation of an intervention programme in complex systems.

The first aim of the LIKE programme is to develop a dynamic action-programme based upon the pre-existing system that shapes health-related behaviours of the population using a participatory approach. The second aim is to evaluate how the system that shapes health behaviours evolves under influence of the programme, and how this relates to changes in the health-related behaviour of adolescents, as well as the prevalence of overweight and obesity. Within these aims, we defined specific research questions relating to intervention development (Questions 1.1 and 1.2) and evaluation (Question 2.1 to Question 4) (see Table 1).

## 2. Theoretical Perspective

### 2.1. Transition from Ten to Teenager

The focus on the transition period from childhood to adolescence (10–14 years) is central to the development of the LIKE programme. This age period is of critical importance in childhood development, with important changes in overweight and obesity as well as health-related behaviours. These changes are related to important biological, psychological and emotional developments in accordance with more autonomy during young adolescence [26]. In LIKE, we aim to examine how these health-related behaviours change over time, as well as the change of related determinants which can help us in defining the most promising mechanisms for intervention, for example, in the transition from primary to secondary school with a focus on ethnically diverse lower socioeconomic subgroups. We address different levels of prevention, including universal prevention (general population, i.e., adolescents in ethnically diverse lower socioeconomic groups), selective prevention (groups within that population that have an increased risk of overweight and obesity, such as children living in poverty), indicated prevention (individuals with overweight and thereby an increased risk of obesity) and healthcare-related prevention (individuals with obesity in the healthcare setting) [27].

### 2.2. Epistemological Pluralism

Second, from the outset of the programme, we purposively applied the viewpoint of ‘epistemological pluralism,’ where we combined the post-positivist and interpretative epistemology to gain a full understanding of the system and system changes [28]. Our programme is post-positivist in the sense that we aim to develop and evaluate the LIKE programme using objective evidence and with prior insights of research on causes of childhood overweight and obesity. But, due to the complex, multiple and unpredictable nature of what is perceived as reality, we believe that an interpretative epistemology is additionally needed with a focus on how different stakeholders perceive the system and the impact of changes in the system [29].

### 2.3. Complexity Thinking

Third, in LIKE, we operationalize complexity by applying a systems dynamics lens throughout the programme development, implementation and evaluation. This implies that we consider the programme to be nonlinear and adaptive where the processes and outcomes are dynamic and cannot be predicted nor controlled in advance (uncertainty) [30,31] and are considered as a complex of actions that changes the system dynamics to result in different emergent properties. We foresee that the potential actions can change the system dynamics in multiple ways following the Intervention Level Framework [32] by changing the: (1) Systems’ mindset or paradigm (e.g., child health is a collective responsibility), (2) the system’s goals (e.g., making the healthy choice the easiest choice), (3) structures (e.g., connecting local growers to local restaurants or schools), (4) feedback loops (e.g., monitoring outcomes of prevention programmes) and (5) structural elements (e.g., availability of healthy food in schools).

### 2.4. Participatory Action Research

Another central theoretical perspective in the LIKE programme is that of a participatory approach. We, in particular, apply the principles of Participatory Action Research (PAR), working both with adolescents and their families, as well as with important stakeholders as co-creators and co-researchers in our programme [33]. This co-design approach aligns with the interpretive epistemology, as well as systems thinking, where we aim to understand the system both from an outside (academic researcher) and inside (adolescent/family/stakeholder) perspective (Table 1). PAR is an iterative process of planning, acting and observing, and is influenced by continuous reflection. By collaborating in the formulation of research questions and designing, implementing and evaluating interventions, the population develops ownership and empowerment [34,35,36].

## 3. Programme Development

### 3.1. Overall Design

Our goal is to develop, implement and evaluate an action-programme at the level of family, school, neighbourhood, healthcare and city as an integrated programme based on a detailed insight of pre-existing systems. Our programme is not defined as a fixed package of activities, but rather as a process of reflection and adaptation as the characteristics of the pre-existing systems become clear to the population and stakeholders [11]. Therefore, the system dynamic and participatory approach can be seen as the intervention rather than the individual actions resulting from this approach. Programme development is adaptive and dynamic using the principles of *developmental evaluation* (see Section 4 below). Ultimately, we aim for the LIKE programme to achieve changes in the systems dynamics, where we do not only see a change in structural elements (for example more healthy school canteens), but also a change of the system as a whole, where health is an emergent property. An overview of the logic model is provided in Figure 1. This study was approved by the institutional Medical Ethics Committee of the Amsterdam UMC, location VUmc on the 15th May 2018. Approval code: 2018.234.

### 3.2. Understanding Pre-Existing Systems (Needs Assessment)

A fundamental part of applying a systems approach is gaining an understanding of the pre-existing system, which is achieved by mapping the system from a post-positivist (researcher-view) and interpretative (stakeholder) perspective. A key challenge when applying systems thinking is identifying relevant boundaries, tracking the consequences of changes in boundaries and determining how different boundaries are linked to each other [31]. The system boundaries in the LIKE programme are set at determinants that we can change (e.g., excluding genetic factors); determinants that are relevant to our population and are related to the target behaviours (diet, physical activity, screen use and sleep) and determinants at the level of family, school, neighbourhood, healthcare and city with a focus on Amsterdam East. We include national and international factors (such as world trade or national policy) to help understand the system but recognize that we are limited in changing these factors within the scope of our programme. We study the pre-existing systems by conducting a rigorous Needs Assessment (NA) using a mixed methods approach. In close collaboration with the target population, data is collected for adolescents with and without obesity, and including families, schools, the health care sector and societal stakeholders (such as retailers, schools or sports clubs). More specifically, in this phase of the programme, we aim to answer two research questions as described below (see Table 1).

Question 1.1. Which factors and processes in the pre-existing system in Amsterdam East shape unhealthy behaviours? (post-positivist).

To answer this question, we aim to develop two systems maps relating to: (1) Relevant behavioural determinants and the every-day lives of our population and their families and (2) the existing networks of stakeholders and ongoing activities of the AHWP.

#### 3.2.1. 10–14-Year-Old Adolescents and Their Families

The LIKE programme not only addresses the adolescents themselves, but also their families, because they shape the contextual environment and are central agents of change in encouraging healthy behaviours [37,38]. To understand the context, we start by building an overview of the pre-existing system from an outside perspective by conducting literature reviews on the most important determinants relating to diet, physical activity, sleep and screen use in adolescents. We summarize our findings in causal loop diagrams (CLDs), where academic researchers review how determinants are connected. This general systems map is specified to our situation in Amsterdam East using in-depth qualitative data that is collected using a range of methods, including observations during activities in our target neighbourhoods (e.g., observations during school lunch breaks, neighbourhood events for adolescents and/or parents), informal conversations with community members and families to build relationships and understand the context of participants daily lives, and photo voice sessions with mothers from the target neighbourhood and in PAR groups (see more details below).

We use the created systems maps in two ways: (1) As a reference mode to develop actions, i.e., actions developed during the programme are checked against the CLDs to make sure we develop adequate actions and (2) as a basis for evaluation. As actions are being implemented throughout the programme, we gain an understanding of the local system and its determinants, and this information is added to the pre-existing systems maps (see more under Section 3 below).

#### 3.2.2. Societal Stakeholders

Alongside the activities with adolescents and their families, we aim to gain a deeper understanding on the influence of societal stakeholders in the system. To do this, we use Social Network Analysis (SNA) [39,40,41,42,43], power mapping (answering the question ‘who holds power in the system?’), action mapping (mapping all actions that are already taking place), and interviews. SNA will be used to study the network of actors in Amsterdam East focusing on all actors working around diet, physical activity, screen use or sleeping and/or have a connection to our target group. From this SNA, we will identify influential actors and invite them to participate in action development. Also, the SNA will form a starting point for action development, as we might see that certain stakeholders are not connected and actions can focus on fostering collaborations between stakeholders [44]. In addition, SNA will be used to detect changes in the network over time by comparing pre- and post-measurements [39,45,46]. Furthermore, we will identify stakeholders who hold the power to facilitate change in the system with the use of power maps. The power maps will be developed alongside the CLDs where we identify who the main actors are who can influence parts of the system. Alongside, we will conduct action mapping with an overview of ongoing activities of the AHWP and their partner organizations within the Amsterdam municipality. This information is essential to ensure that we do not develop actions that are already taking place and to identify important stakeholders to collaborate with. Finally, we will conduct qualitative interviews with professionals in the health care sector to gain an understanding on barriers and facilitators in achieving optimal support and empowerment for youth with obesity.

Question 1.2. From the perspective of adolescents/families/societal stakeholders, which factors and processes in the pre-existing system in Amsterdam East shape unhealthy behaviours? (interpretative).

A crucial step to achieve systems changes is building a model of the system as viewed by adolescents, their families and societal stakeholders [47]. Methods to answer Question 1.1 provide in depth qualitative and quantitative information to create a post-positivist view of the system. In the current step, we supplement this view with interpretative system models developed during the PAR groups and photo voice interviews (adolescents) and Group Model Building workshops (societal stakeholders).

The PAR groups include 6–8 adolescents per group and 1–2 facilitating academic researchers. PAR groups are set up for adolescents in a school setting and specifically for adolescents with obesity in the health care setting. PAR groups meet weekly throughout a predefined period, and in these meetings, they conduct research among their peers on their needs regarding healthy behaviours and potential actions toward stimulating healthy behaviours. Through capacity-building workshops, adolescents learn basic research principles and methods that enable them to conduct research. Furthermore, adolescents and families are encouraged to develop an individual and collective view on their current lifestyle using photo voice. This is facilitated in several sessions where participants take photos, collect and select images, discuss and reflect on the images and bring about action. Each PAR group summarizes their research results in a CLD.

With the societal stakeholders, we use Group Model Building, which is a participatory approach to build capacity to think in systems. This approach has been found to improve problem understanding, increase engagement in systems thinking, build confidence in the use of systems ideas and develop consensus for action among diverse stakeholders [48,49,50]. We conduct Group Model Building workshops with societal stakeholders who hold a central position within the local governance and/or at community level. They are identified through the SNA, power maps and action maps, as well as through a systematic selection process by the AHWP, and represent the following sectors: Schools, sport clubs, local government, parents, community workers, retail and health care.

### 3.3. Developing Quick and Disruptive Actions

Building on the understanding of the pre-existing system, the next part of LIKE focuses on developing the action-programme. As mentioned before, the LIKE programme will be a complex programme, meaning that it includes non-standardized and adaptive actions, ‘emergent’ in response to changing needs and understanding of what is working. We aim to work on actions that can change elements within the system, as well as actions trying to change the system as a whole [47,51]. We do this by developing different types of actions, including quick actions (quick testing, adapting and possibly catalysing further action) and more disruptive actions (aiming at paradigm change or the goals of the system). The co-created action ideas will concentrate on the *function* (e.g., improving healthy food access) rather than the *form* (e.g., free fruit in schools), which is in line with our systems lens where interventions are seen as events in systems and not as a package of activities [11]. To ensure action is taking place at different levels, actions are checked against the Intervention Level Framework [32] and against the CLDs.

#### 3.3.1. Quick Actions Targeting System Elements

Quick actions are actions that can be implemented relatively quickly, following principles of learning by doing. By observing and monitoring, the working of these actions can be used to further increase the knowledge of the system and consequently stimulate larger actions. The general principle is that there is not one silver bullet to address childhood overweight and obesity and that we need many silver bullets to be fired at the same time. Here, we might see an accelerating effect. In systems terminology, this principle is referred to as nonlinearity, where small actions can stimulate larger actions [31]. Despite being “quick,” it is important that the actions are robust. We therefore developed a stepwise checklist for each action, including checking each action with the pre-existing systems maps and existing initiatives.

Within the PAR groups, adolescents work as co-creators and together with researchers to develop and implement their own action ideas where they work in subgroups on diet, physical activity, sleep and screen use, supported by the AHWP. Within the Group Model Building sessions, once the societal stakeholders develop a shared understanding of the system, we identify where action is already taking place (from the action maps) in the system, and by whom (from the power maps and SNA). Following this activity, stakeholders identify the most important feedback loops and thereby areas of priority for action. These priority areas are then developed into concrete action plans where we work with the same stakeholders from the Group Model Building workshops supplemented with others who have the power to make changes. We aim to form a maintainable stakeholder network that meets regularly and continues to exist after the LIKE programme ends [52].

#### 3.3.2. Disruptive Actions

We pay specific attention to achieving not only changes in systems’ elements, but also at the level of system goals or paradigm. To achieve this, it is important to identify the system dynamics and underlying mechanisms we aim to disrupt [16]. We aim to do this by combining the insights from the various methods described above and use this information to discover mechanisms and leverage points for change using the aforementioned Intervention Level Framework [32]. The goal is to provide examples of actions at each level and to describe actions in terms of goals/function (e.g., improve the network structures between parents and schools). This way, adolescents and societal stakeholders will be encouraged to develop actions at different levels, including more hard-to-reach goals of paradigm change. Here, we use methods of action plan development such as the ANGELO framework [53]. Mechanisms, leverage points and action ideas will be co-developed with the AHWP to find ways to implement ideas in existing initiatives or policy or, where needed, to develop something new. The AHWP has reserved budget for these actions, which can also be implemented after the LIKE programme has finished and can be integrated in wider AHWP or city council activities.

## 4. Evaluation

### 4.1. Developmental Systems Evaluation

An evaluation study accompanying interventions in complex systems can serve multiple purposes depending on the phase of the programme [54]. At the start of the LIKE programme, our evaluation aimed to support the development and implementation of the actions by: developing an understanding of the system; framing and co-creating actions that can change the system, supporting sustainable implementation of the co-created actions and in assessing the potential value of the actions and overall system dynamics and participatory-based approach. The second aim of our evaluation was the production of generalizable knowledge, both in terms of outcome and process, to be used in other contexts at national or international level.

*Developmental evaluation* is particularly well suited for these purposes, as it includes principles of adaptation, ongoing programme development and feedback on broad systems change [31,55]. More specifically, in contrast to more traditional evaluations where control and prediction are desirable, developmental evaluation is suitable for interventions trying to target complex environments characterized by high uncertainty. It tries to identify and explain patterns of change that emerge as the intervention unfolds [31]. We developed an evaluation framework informed by key principles of developmental evaluation and building on recent work by Moore et al. [16], Walton [25] and Egan et al. [56], resulting in a developmental systems evaluation framework. Table 1 provides an overview of our evaluation framework (summarized in Figure 1), including the evaluation questions, related systems principles and boundaries, and specific methods to answer each question.

### 4.2. Evaluation to Support Action Development

Question 2.1: How does the action-programme evolve and of which action elements does it consist?

By the nature of our programme, we do not know in advance what our actions will look like and to what extent they are targeting different system levels. Therefore, this evaluation question relates to the types of actions that are developed during the LIKE programme and where they fit on the Intervention Level Framework. We will monitor all actions that are being developed, who is initiating the actions and whether they take place at the level of adolescents, families, school, neighbourhood, health care, city or beyond. We keep track of programme development by keeping detailed logbooks of all PAR and Group Model Building meetings and by keeping notes of all LIKE consortium and stakeholder meetings. Implementation is guided through an extensive steering model including numerous steering committees with a mix of academic researchers and the AHWP, including the Consortium Leaders (meeting every six weeks); Management Board (all senior members meeting four times per year); Consortium Meeting (all members, meeting four times per year) and two-weekly implementation meetings (junior researchers with AHPW staff). We also conduct interviews with various participants and stakeholders during different stages of the programme. Furthermore, we keep track of barriers and facilitators for implementation of actions, including, for example, lobbying activities that are required to achieve policy change. Finally, we will specifically monitor whether we see actions arising at higher system levels (goals of the system or paradigm change) as the programme unfolds. This information will also be used to adapt the programme activities where needed.

Question 2.2: How successful is the approach we follow in our LIKE programme in creating a sustainable programme and how can this be optimized?

In LIKE, we propose a combination of systems dynamics and participatory methods to support intervention development including Group Model Building and PAR. Our hypothesis is that by involving adolescents and their families, as well as societal stakeholders, and learning and stimulate them to think in systems; we will develop actions with a potentially larger and more sustainable impact. To know whether these methods are effective, it is important to know how well they were implemented. Therefore, we aim to conduct a process evaluation including questions on the function rather than the form of the actions, e.g., did the use of the participatory approach lead to a better reach of the adolescents? Were we able to engage a wide range of societal stakeholders within the Group Model Building process? This will be evaluated as the intervention development evolves. During this time, we can make alterations to our methods, for example, by inviting new stakeholders or increasing the involvement of school staff.

### 4.3. Evaluation to Examine Systems Changes

Question 3.1: What type of (emergent, adaptive, reinforcing) changes occurred in the living context, what were potential unintended consequences and how can these be related to the LIKE programme (post-positivist)?

Methods that can be used to understand changes in interconnections in the system include Group Model Building, CLDs and SNA [51]. All actions that follow from the LIKE programme will be integrated in the CLDs that were developed as part of the needs assessment and during the Group Model Building workshops. This way, we can measure if and how the actions cause changes to the system. We estimate whether the intervention changed the target determinant (i.e., healthy food access), the target behaviour (i.e., junk food consumption), whether it was implemented as intended (function), whether it caused unintended consequences and how it relates to other determinants in the system. We will monitor: (1) The CLDs as they evolve during the programme and (2) system-level changes according to the Intervention Level Framework. Based on these measures, we will update the CLDs accordingly in order to identify new points for action or adapt actions to achieve a more impact. In addition to the CLDs, we will monitor how local networks change over time by conducting SNA at multiple time points and comparing connections and the strength of different relationships.

Within our programme, it is not feasible to conduct the analyses described above (i.e., mapping systems over time) for a control region. Our empirical basis for judgements about causality are therefore not a traditional comparison between intervention and control. Instead, we aim to capture a comprehensive insight into the system as a whole (see Question 3.2 below), supplemented with a quasi-experimental sub-study to capture elements of attribution for specific determinants (Question 4).

Question 3.2: How do the target group and stakeholders perceive changes in the system and how do they perceive the contribution of activities within the LIKE programme to these changes? (interpretative).

Based on a paper by Ling et al. [57], this part of the evaluation does not aim to identify attribution (what proportion of the outcomes was produced by the intervention?) but instead to clarify contribution (how reasonable is it to believe that the intervention contributes to the intended goals and might there be better ways of doing this?). For example, rather than asking ‘what factors need to be present in order for this intervention to work?’, we ask ‘how do the factors interact with each other, how do these interactions change over time and to what extent are these amenable to intentional change?’.

We will measure this concept using ‘Contribution Stories’. These stories capture ideas from the adolescents, their families and societal stakeholders about how the different activities interact with each other and with other systems. From these participants’ Contribution ‘Stories,’ more abstract Theories of Change can be developed which trace the causal pathways linking resources used to outcomes achieved [57]. We will use our CLDs as a starting point for conversation in these contribution stories, where participants will be able to update the CLDs based on the perceived intervention impact.

Question 4.1: To what extent do adolescents’ behaviours and weight status improve as a result of changes in the system?

Adolescents in Amsterdam are systematically monitored from birth, throughout their school careers, until the age of 18 years. Data collection includes digital client files (standardised), questionnaires, and medical examination. The use of these routinely collected data provides the opportunity to monitor medium to long-term trends in the ultimate outcomes of the LIKE programme using a quasi-experimental design. For Body Mass Index (BMI z-score; relative weight adjusted for child age and sex), we will use BMI measured during the routine child health care consultation at the age of 10 and 13–14 years. Diet, physical activity, screen use and sleep are also routinely measured at these ages. For all outcomes, we will be able to use repeated cross-sectional data, where the main analysis will focus on time-trend analysis, examining the change in trends of health behaviours/BMI in intervention versus control communities.

The LIKE programme was implemented in three communities in Amsterdam East. Using a quasi-experimental design, we matched control communities to these intervention communities based on demographic characteristics of the population. Here, we will use two types of comparison districts in Amsterdam: Those in which the AHWP has been implemented, to study the additional effect of the LIKE programme, and those without AHWP, to study the effect of the LIKE alone. It is important to note that the total lead time of five years for the evaluation study might be too short to establish conclusive evidence on the impact on ultimate outcomes. For example, it may take some years before changes in health behaviours result into a reduced prevalence of overweight or obesity at the population level. Therefore, this part of the evaluation will run in parallel with the other evaluation questions (see Figure 1), where, in due time, we might be able to see at what point system changes (in the CLDs/SNA) also result in measurable changes in health behaviour and/or weight status.

Question 4.2: How do we expect the observed changes in overweight and obesity in adolescents to translate to incidence of cardiovascular disease in adulthood?

Finally, to quantify the effect of changes in overweight and obesity in adolescents on incidence of cardiovascular disease (CVD) during adult life (20–70 years), we used the DYNAMO-HIA model (http://www.dynamo-hia.eu/), which is a dynamic modelling tool for quantifying effects of changes in risk factors on health outcomes. Repeated measurements of BMI described above, as well as insights obtained from other cohort studies, will be used to estimate the tracking of (changed) overweight and obesity at childhood ages into overweight and obesity at the start of adulthood. The DYNAMO-HIA tool will next be used to quantify incidence of CVD in the reference scenario (no intervention) and intervention scenario during adult life and to perform sensitivity analyses.

## 5. Discussion

This paper introduced the LIKE programme, which follows a systems dynamic and participatory based approach to promote healthy habits among 10–14-year-olds in multi-ethnic, lower socioeconomic groups in Amsterdam. We described the way we aim to co-create a dynamic action-programme based on a continuing process of reflection and adaptation as the characteristics of the complex systems become clear to the population and societal stakeholders. The development of the action-programme will be accompanied with a developmental system evaluation to support its development and assesses how the programme leads to changes in the system and the behaviour itself. Some strengths and limitations need to be discussed, concerning both the development and the evaluation of LIKE.

As stated in the introduction, the systems lens we apply goes beyond implementing interventions at multiple levels and adds the recognition of each level as systems that follow principles of system dynamics including linkages, relationships, feedback loops and interactions. Based on this thinking, we aim to develop actions that change the system at different levels, from a shift in paradigm to changing structural elements of the system. Previous reviews have shown that current interventions in public health are mostly concentrated at the level of targeting structural elements [32]. Likewise, a recent study using Group Model Building in New Zealand communities showed that while it was possible for stakeholders to think in systems, they were unable to identify solutions beyond structural elements [58]. To navigate this risk, we place specific emphasis on achieving changes at higher system levels throughout the action-programme development using the Intervention Level Framework.

Related to the previous issue, we realize that our programme is highly ambitious. We therefore made substantial effort to create the conditions that will enable us to accomplish these ambitions. This includes the formation of an interdisciplinary consortium of scientists and societal partners, the latter embedded within the AHWP. The AHWP, in itself, provides an exceptional opportunity to achieve the ambitions of the LIKE programme, facilitated by the unique combination of conditions created over the past years, including strong political support, commitment and involvement of community organisations in the objectives of the programme, and involvement of public sectors other than public health (e.g., education, spatial planning) [59]. This means that we do not have to start from scratch in creating community or policy involvement and that a lot of the networks and structures already exist. Collaboration with so many different stakeholders also raises its own problems, however, in relation to roles, expectations and results. We navigate these tensions by having regular meetings where such topics are openly discussed and by trying to build trust between the different partners.

Other challenges of the LIKE programme relate to the evaluation, which should: (a) Support the development of the programme locally, (b) produce results that can be used for the development of similar programmes in other settings and (c) mitigate the scientific and political risk to conclude that the programme is not effective although it produced important changes at the community or organizational level [28,60]. We propose a developmental systems evaluation design. This design will not result in a fixed intervention, but in an adaptive action-programme that works in tandem with our growing knowledge of the pre-existing systems. We believe this approach is key in developing an impactful and sustainable programme, but it complicates generating knowledge that is generalizable to other settings. We propose to address this by combining qualitative and quantitative data on systems changes and detailed measures of contribution. In parallel, we aim to assess changes in health behaviour and body weight that result from the programme and interpret these outcomes in tandem with the systems changes. This way, we can monitor when system changes translate to changes in healthy behaviour and body weight, although the latter might require a longer evaluation timeframe.

To the best of our knowledge, there are only a few studies that aimed to evaluate how the system that shapes individual behaviours changes under influence of intervention programmes [61]. Recent work by Public Health England has shown how whole-of-systems approaches to obesity can be effective to help communities think in systems, but has not yet been able to examine how this way of working impacts on the local system of obesity causes [62]. Hennessy et al. developed retrospective systems maps representing community change dynamics within the Shape Up Somerville (SUS) intervention [61]. We propose to prospectively measure systems changes by conceptually linking the pre-existing systems maps with the developed actions and tracking changes over time in different levels of the system elements, system structures and/or the system as a whole, similar to that proposed by Egan et al. [56]. Therefore, data collected during the needs assessment phase will not only be used as a basis for developing actions, but will also serve as a ‘baseline’ measurement. We expect that the use of mixed methods, combining both quantitative and qualitative data, will provide us with an in-depth analysis of system changes both from the perspective from academic researchers, as well as the adolescents, their families and societal stakeholders.

## 6. Conclusions

This paper presented the design of the LIKE programme, which combines a systems dynamics and participatory approach aimed at tackling the complex problem of childhood overweight and obesity. The methods described can serve as a case study for the development and evaluation of interventions in complex systems. In the years to come, we will report the effectiveness of our programme, as well as whether our evaluation design enabled us to understand why it worked and for whom. We realize that the proof of the pudding is in the eating, but we nevertheless believe that our methodology can serve as a source of inspiration for future evaluation studies accompanying complex interventions.

## Figures and Tables

**Figure 1 ijerph-17-04928-f001:**
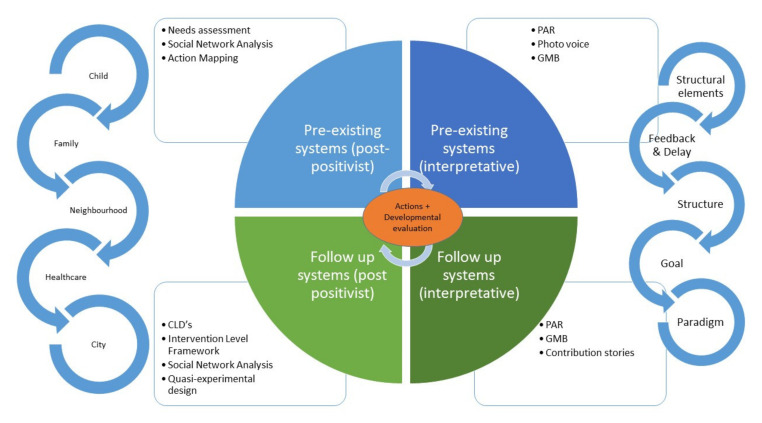
Overview of the different components of the LIKE programme and how they relate to each other. PAR = Participatory Action Research. GMB = Group Model Building. CLD = Causal Loop Diagram.

**Table 1 ijerph-17-04928-t001:** An overview of the research questions, design and methodology of the LIKE developmental evaluation.

Evaluation Question	Population	Epistemology		Outcome Measures	Data Collection	Systems Principles (Based on [24,25])
		Post-Positivist	Inter-Pretative			
**1. Understanding the pre-existing system in relation to health-related behaviours** 1.1 Which factors and processes in the pre-existing system in Amsterdam East shape unhealthy behaviours?	Adolescents and their families	V		Pre-existing system of relevant health behaviours and determinants and how they are connected	Literature reviews Participant observations PAR groups—determinant research Action mapping of ongoing activities Combining all data listed above into overarching CLDs (pre-existing system in Amsterdam East)	Developing an understanding of the system, information from different agents, participatory methods. Question provides input for programme development
	Stakeholders	V		Baseline SNA	Social network analysis (baseline): mapping all the local actors who are involved with the target behaviour and are in touch with the target group and the connection between them	Information from different agents Relationship between agents
	Adolescents with obesity and their families	V	V	Barriers and facilitators of support and empowerment towards sustainable healthy habits	Interviews with healthcare professionals Participant observations Observations during health visits Go-along interviews	Information from different agents, participatory methods.
1.2 From the perspective of adolescents/families/societal stakeholders, which factors and processes in the pre-existing system in Amsterdam East shape unhealthy behaviours?	Adolescents and their families		V	Pre-existing system/CLD from target group perspective	PAR groups CLD building Photo voice	Developing an understanding of the system Participatory methods Question provides input for programme development
	Societal stakeholders		V	Pre-existing system/CLD from stakeholder perspective	Group Model Building with local stakeholders	Developing an understanding of the system
	Adolescents with obesity and their families		V	Needs for lifestyle behaviour change by adolescents with obesity and their parents	PAR groups with adolescents with obesity and/or their parents	Developing an understanding of the system Participatory methods
**2. Developmental evaluation (process)** 2.1 How does the action-programme evolve and of which action elements does it consist?	Adolescents and their families	V		Overview of activities listed by Intervention Level Framework Barriers & Facilitators	PAR groups Logbooks Notes Meetings with implementation partners	Developmental evaluation Dynamic Adaptation Feedback
2.2 How successful is the approach we follow in our LIKE programme in creating a sustainable programme and how can this be optimized?	Adolescents and their families	V	V	Process evaluation	Reach and implementation of the PAR groups Reach and implementation of the GMB workshops Levels of systems thinking by participants Function of the GMB and PAR groups (did it lead to different types of actions?)	Developmental evaluation Dynamic Adaptation Feedback
**3. Developmental evaluation (effect)** 3.1 What type of (emergent, adaptive, reinforcing) changes occurred in the living context, what were potential unintended consequences, and how can these be related to the LIKE programme?	Adolescents and their families	V		Adapted CLDs Extent to which the adapted system is “healthier”	Actions that were implemented during the programme and plotting these on the baseline system maps (CLDs) and against Intervention Level Framework. Feedback in the baseline systems following these actions. This includes: Potential intended and un-unintended consequences Collecting relevant data (depending on the exact actions that are being implemented) Add to the CLD	Adaptation Emergence Non-linearity Case study design Multiple and mixed methods
	Societal stakeholders	V		Follow up SNA	Social network analysis (follow up): mapping all local actors who are involved with the target behaviour and are in touch with the target group and the connection between them	Information from different agentsRelationship between agents
3.2 How do the target group and stakeholders perceive changes in the system and how do they perceive the contribution of activities within the LIKE programme to these changes?			V	Perceived changes Contribution (how reasonable is it to believe that the programme contributed to the observed changes?)	Contribution stories/sensemaking using interviews with stakeholders and target group based on the CLDs PAR groups	Information from different agents Contribution Participatory methods
4. To what extent do adolescents’ behaviours and weight status improve as a result of changes in the system?	Adolescents and their families	V		Routinely collected data on BMI; diet, sleep, PA, screen time	Quasi-experimental design	Time (delay) Emergence

BMI = Body Mass Index, CLD = Causal Loop Diagram, GMB = Group Model Building, PAR = Participatory Action Research, SNA = Social Network Analysis.

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
