# Peer review of "A System Dynamics and Participatory Action Research Approach to Promote Healthy Living and a Healthy Weight among 10–14-Year-Old Adolescents in Amsterdam: The LIKE Programme"

_ijerph, 2020, doi:10.3390/ijerph17144928_

Round 1

Reviewer 1 Report

This manuscript described the design of a large study and aimed to at utilizing a systems dynamic and participatory based program to reduce obesity and adolescents in the Netherlands. The authors provided a comprehensive and detailed description of the study design and its potential impacts at various levels of the system (e.g., stakeholders, adolescents and their families). There is much enthusiasm for the project and the manuscript, which was well written. There are only a few suggestions for improvement:  

  1. There were many acronyms used throughout the paper. I recommend reducing the number of acronyms to improve readability.

  1. Overall, the manuscript could be shortened, although comprehensive and well written, the manuscript was quite lengthy.

  1. Further, as written, it was unclear, at times, how this study and approach will target weight specifically. The authors might include examples throughout that directly state how this approach will impact weight, obesity, and other health-related behaviors, specifically. For example, how might the program engage with retailers, schools, sports clubs, government, community workers, etc. to enact change in the system and what quick actions might the study involve. Although the exact mechanisms will be identified as part of the study, some expected examples might clarify how weight, as opposed to other public health concerns, might be targeted in this program.

See below for a few minor suggestions:

  • Line 63. The authors might include an example of how change at the environmental level doesn’t render the expected results.

  • Line 80, the authors might add examples of linkages, relationships, feedback loops, etc. to help the reader fully understand what these processes look like in the real world.

  • Line 126. There is a small typo – a space is missing between being and developed.

  • Table 1. This table was extremely helpful, however, the formatting made it hard to discern which box some of the text belonged to. For example, the lack of space between rows makes it hard to know if the text in the data collection and systems principles columns belong to the adolescents or stakeholders under evaluation question 1.

  • Figure 1. Could the authors either spell out abbreviations in the figure or add a footnote clarifying these abbreviations.

Reviewer 2 Report

This document presents the design of the LIKE program that combines a systems dynamics and a participatory approach, aimed at addressing the complex problem of childhood overweight and obesity. The topic addressed is highly relevant internationally and an innovative approach is presented, it is very interesting to wait for the results obtained from this study.
I suggest that the authors emphasize the strengths and possible weaknesses that could arise.

Reviewer 3 Report

Thank you for giving me the chance to review the manuscript ijerph-846633 entitled "A systems dynamics and participatory action approach to promote healthy habits and a healthy weight amongst 10-14 years olds in Amsterdam: Design of the LIKE programme” in consideration for publication in the International Journal of Environmental Research and Public Health.

The aim of the manuscript is to outline the design of the innovative LIKE programme. This aim has been achieved by authors and this altogether described a valuable and interesting study programme. The manuscript is well-written, easy to follow and clearly described with good use of a summarizing table and a figure.

The authors have stated a clear case for the study and thus provided a well-described rationale for the undertaken programme. The specific study questions are outlined and described in detail.

Finally, the authors have stated that they hope that the study design serves as a source of inspiration for other researchers. As a fellow researcher in the field, I have to say that reading about this programme has been very interesting and I am eager to hear about the process and findings of this planned piece of work. Thank you therefore for this opportunity to review this paper.

After reading about the design of the programme, I was left wondering how this would be assessed and documented through the process, in order to assess whether the participatory action research was achieved, and how this was implemented in the different stages. I was further wondering if a scientific advisory board or other stakeholders or also a public engagement board were involved at all in the study. If so, this could be stated as well to explain to the reader how the process was overseen and evaluated.

This work, in the future, could be added by a more precise study protocol, amongst others, to provide further information on the following: indicating the planned questionnaires and measures, methods, number of participants, recruitment and the planned analysis. Also, I was left wondering how participants will be accessed and recruited, who the network players and stakeholders that were referred to are, in how far schools are involved, how co-creation would be done with multi-ethnic groups and whether it was considered to avoid possible stigmatization amongst the target group. Only an outline of these aspects has been provided within the present manuscript which leaves some questions open. This information could alternatively also be stated the study protocol.

I would like to raise the following points which might help to further improve the manuscript:

Title:

  • The title was reflective of the aim and the content of the paper. In my opinion the title should state: “A system dynamics and participatory action research approach to promote healthy habits and a healthy weight among 10-14 year olds in Amsterdam….”
  • It could help to indicate ‘adolescents’ or ‘young people’ in the title in order for it to be found and detected in other searches, but this is not a necessity.

Abstract:

  • The abstract effectively summarizing the manuscript. It did not become clear if the programme was going to be implemented over 5 or 10 years (Lines 44-46). This could be made clearer in the text to avoid confusion.

Background:

  • The introduction was succinct and well structured, and the approach taken in the study was justified.
  • The AHWP programme is introduced in the aims and objectives section. It might be worthwhile to provide a bit more information about this programme, how long it has been running, etc. Is there a reference that could be added?
  • Table 1: Whilst the idea of providing the content within a table is good, the table, as it currently is presented, it is fairly hard to read and follow. It could be improved by changing the size of the font, providing grey backgrounds to differentiate between lines, by explaining what ‘V’ stands for, and by explaining the abbreviations used in the table (such as SNA).
  • Table 1: The numbers of questions in the table is incorrect (“developmental evaluation” named as 1, sub question named as 2.1) so should be re-worked.

Programme development:

  • Figure works well but includes many abbreviations which were not previously explained in the text. (for instance, GMB)
  • Line 197: I believe you mean to refer to section 4, not 3.
  • Line 247: What do you understand and mean by power mapping, action mapping? It would be helpful to provide references at this stage.

General comments

  • Some minor grammatical issues, type-editing issues. All abbreviations should be checked. They should be mentioned once in full, and then used thereafter in the abbreviated version.
  • In some references the year is mentioned (page 13, lines 347-348) which is not necessary.

Reviewer 4 Report

The present study presents the methodology of the LIKE program, an integrated action program utilizing systems thinking and participatory action research to overcome many limitations of prior intervention and prevention studies in pediatric obesity. Although the project is highly ambitious, it is a novel and important test of systems thinking and PAR in childhood obesity. As currently written, the manuscript highlights many important areas, however due to the complexity of the project, I feel as though the manuscript could benefit from some clarification and streamlining.

Generally, the manuscript contains a vast amount of information related to the project. The authors should consider streamlining the manuscript (e.g., removing redundancy from introduction and discussion) and highlighting key areas in order to direct the reader’s attention.

Within the introduction it would be helpful to focus more on the population of interest. Specifically, this intervention targets a key area of childhood in a racially/socio-economically diverse sample, and this context is important to consider more clearly in how the age of the population may impact the PAR and systems thinking.

Additionally, clarity about the primary goal (i.e., intervention vs. prevention) and rationale would be helpful.

Table 1 would benefit from streamlining in order to aid use of the table. For example, removing the epistemology and level/boundary columns may help increase focus on key aspects of the evaluation.

How will the multiple causal loop diagrams be integrated?

Please provide more information about the feasibility of the quick and disruptive actions, especially within the context of suggestions provided. For example, if it is suggested that a better grocery store be available within walking distance of families’ homes, how will that be acted upon? Providing examples of what some of the direct actions that the project might take may help highlight the potential changes occurring the in community.

The evaluation components could be clearer, especially regarding how you will use the evaluation components to answer the research questions. For example, how will you measure that you have engaged a wide range of stakeholders? (page 14)

As the AHWP is a key aspect of the program mentioned throughout the manuscript, it would be helpful to have more context for what the AHWP is and how the proposed project is adding to the existing AHWP.

I would appreciate brief expansion on how this project is actually feasible, given the discussion of this point at a limitation, but lack of clarity surrounding how this is overcome within the current research strategy.  
